# Developments and Applications of Artificial Intelligence in Music Education

**Xiaofei Yu [1], Ning Ma [2], Lei Zheng [3], Licheng Wang [4] and Kai Wang [2,*]**

[1] Conservatory of Music, Qingdao University, Qingdao 266000, China
[2] School of Electrical Engineering, Weihai Innovation Research Institute, Qingdao University, Qingdao 266000, China
[3] Science and Technology Department, Qingdao University, Qingdao 266000, China
[4] School of Information Engineering, Zhejiang University of Technology, Hangzhou 310014, China
[*] Correspondence: wangkai@qdu.edu.cn or wkwj888@163.com; Tel.: +86-15863060145; Fax: +86-532-85951980

**Abstract:** With the continuous developments of information technology, advanced computer technology and information technology have been promoted and used in the field of music. As one of the products of the rapid development of information technology, Artificial Intelligence (AI) involves many interdisciplinary subjects, adding new elements to music education. By analyzing the advantages of AI in music education, this paper systematically summarizes the application of AI in music education and discusses the development prospects of AI in music education. With the aid of AI, the combination of intelligent technology and on-site teaching solves the lack of individuation in the traditional mode and enhances students' interest in learning.

**Keywords:** artificial intelligence; music education; applications; developments

## 1. Introduction

Artificial Intelligence (AI) is a new technical science to research and develop the theory, method, technology and application system for simulating and expanding human intelligence [1–5]. It is a branch of computer science, involving philosophy, cognitive science, mathematics, neurophysiology, psychology and a range of other disciplines. It is also a challenging subject [6–10]. Since AI was formally put forward in 1956, AI has experienced more than 50 years of development and become an interdisciplinary and frontier science.

The emergence of computers has promoted the development of modern electronic music technology. With the rapid development of computer multimedia technology and signal processing technology and its penetration into the field of music appreciation and creation, modern music technology represented by electronic music has developed rapidly, and the field of technological innovation is gradually expanding [11–13]. When it comes to the application of AI in the field of music, music technology has to be mentioned [14,15]. Music technology is an interdisciplinary subject, divided into art and technology parts [16–18]. The art part mainly studies the use of various audio softwares for music creation and production; the science and technology part mainly studies the use of computer technology to provide technical support for music production. Figure 1 shows the relationship between AI and music education. Due to the combination and development of music education, AI technology has become the future trend of music education, exerting a huge influence on traditional teaching concepts and methods and forming a diversified and multi-level development direction. In recent years, digital music has become a huge part of the music industry [19,20]. The combination of audio big data and AI generates Music Information Retrieval (MIR), which is based on music acoustics and extracts audio features based on audio signal processing. Various machine learning technologies in AI are widely used at the back end, which is the most important part of music technology. With MIR, we can use

music as a kind of information to carry out information retrieval, so that we can classify the huge music library and conduct more detailed study on the elements of music, such as pitch and rhythm. In addition to MIR, current music technologies include AI composition [21], song synthesis technology [22] and digital audio watermarking technology [23]. Although these music technologies are not perfect and have certain limitations, they have played a particular role in promoting the development of the music industry and have their own theoretical and practical value, which will be widely used after continuous improvement in the future [24–26].

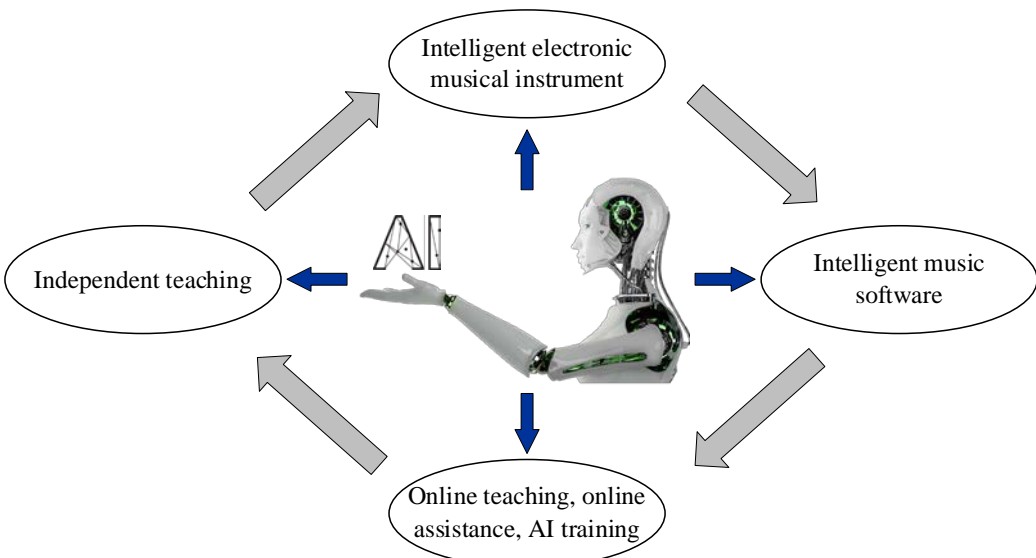

**Figure 1.** The Relationship between AI and music education.

The organic integration of AI technology and music education has enriched classroom teaching resources, expanded the functions of intelligent instruments and improved the technical means of music education. It supports personalized learning, analyzes the melody and rhythm of music, effectively evaluates the teaching effect, and inspires music teachers to use artificial intelligence technology to innovate music teaching. This paper systematically summarizes the application of AI in music education, including the application of AI in intelligent electronic instruments, intelligent music software, online teaching and autonomous teaching. In the rest of the paper, Section 2 discusses the application of combining AI and music education, Section 3 discusses the development, significance, and prospect of AI and music education.

## 2. The Application of Combining AI and Music Education

### 2.1. Application in Intelligent Electronic Musical Instruments

In recent years, the continuous development of AI technology has made electronic musical instruments more intelligent, humanized and specialized to bring forth the new [27]. The intelligent electronic instrument can not only store all kinds of musical instrument timbre, but also realize the effective combination of all kinds of timbre, so that all kinds of timbre can be performed according to different action instructions. This function is obviously difficult to achieve for traditional musical instruments. With these advantages, intelligent electronic instruments are gradually introduced into music teaching to guide students to learn new intelligent electronic instruments. It is because of the introduction of intelligent electronic musical instruments for music education, that a new mode of education is provided. More than ever, one person alone can play, and through various combinations of effective sounds, expand creative thinking. Music provides great convenience for the students of music practice, and further gain a higher quality of teaching [28–32].

Xu et al. [33] studied the collaboration between electronic music creation and online performance of music education and wireless networks under AI. Today, with the rapid development of science and technology, AI technology continues to progress, and the development of digital technology, electronic music online performance, and wireless network collaborative research is more important. Through the research of AI, an electronic music creation system was designed, which realizes the cooperation between electronic music online education and wireless networks. The concept and technology of computer sensor networks, intelligent algorithms and wireless networks are studied, and it becomes a new type of intelligent electronic musical instrument. Through the simulation experiment, the matching degree of AI electronic music course resources are verified, and the oscilloscope is used to transform the sound characteristics into the corresponding sound and image patterns, so as to achieve the purpose of online electronic music intelligent matching and to realize the function of online education.

Guo et al. [34] proposed a new method of piano teaching. Under the framework of an AI environment and wireless network optimization, they adopted a new piano teaching method of "people + equipment", and constantly improved two piano teaching modes: "complementary" piano teaching mode and "remote network" piano teaching mode, which conforms to the trend of the integration of piano performance form and current high-tech development. The function and role of AI is reflected in intelligent teaching, intelligent scoring, networked piano classrooms, and automatic playing functions. The combination of traditional piano teaching and modern AI technology innovation promotes the renewal of new piano education concepts, the continuous advancement of the piano education industry, the continuous improvement of the system's power, and gradually improves the standardization and specialization of the piano education industry. Liu et al. proposed the design of an intelligent piano performance system based on AI and studied the realization of the piano teaching system. The teaching system from the angle of the simulation of the teachers, based on the piano teaching evaluation system was put forward. For the system as a whole, the function of the piano, including signal extraction, play interface, etc., through the experiment verified the feasibility of the teaching system. The system also can simulate teachers guiding students to play the piano, which is of great significance. By using this kind of software, students can detect the music they play in the process of piano practice at home, and more intuitively, see the problems in their playing mistakes. For some common mistakes, they can solve them directly without bringing them to the classroom. At the same time, for some mistakes, specific and correct playing positions will be provided on these software detection pages, which can help students to quickly find them on the piano, greatly improving the efficiency of piano practice and the quality of the class. At present, intelligent electronic instruments have been favored by consumers in the market, such as the intelligent electronic piano. Compared with the complex and expensive traditional piano, the intelligent electronic piano is cheap and easy to use. At the same time, it is also equipped with a self-study software app, which is more suitable for self-study at home.

The comparison between traditional instruments and intelligent electronic instruments with AI is shown in Table 1.

**Table 1.** Comparison between traditional instruments and intelligent electronic instruments with AI.

| Traditional Instruments | Intelligent Electronic Instrument with AI |
| --- | --- |
| Need relatively solid basic skills | Assist the performer to complete the music performance and reduce the difficulty of performance |
| One person only can play the instrument | One person can play multiple instruments |
| No such function | Realizes the cooperation between electronic music online education and the wireless network |
| No such function | Intelligent teaching, intelligent scoring |

## 2.2. Application of Intelligent Music Software

The application of AI music software depends on the output of electronic equipment and whether the processing ability of music data has been restricted by conditions, however, the storage of music information is more stable. Users can edit, adjust, record freely, and process various music elements with AI. With the popularity of music teaching, AI music software provides an interactive platform for teachers and students to share learning resources, where teachers or students can find their own resources to improve. The traditional way of music teaching has undergone a huge change. The knowledge that the teacher teaches in the music teaching class and the content that the student is interested in expanding can be completed by AI music software. Advanced music software includes all kinds of music elements, which broadens students' music vision and deepens their music perception. At the same time as spreading the charm of music elements, it can provide a platform for teachers and students to communicate with each other, or leave feedback or play together, so that music teaching class is no longer limited to the interaction of imparting and absorbing, presenting a positive communication between teachers and students [35–37].

Zhao et al. [38] through the combination of AI and professional platform analysis, evaluated the changes in education and teaching with the coming of the AI era, and put forward alternative topics for music education ability and development. They first discuss the key link between the ability and development of music teachers in primary schools, and demonstrate a sound system and teaching environment of music education in primary schools in the era of AI. Summarizing the experience in practical education provides an important reference for promoting the development of students' personality and ability, and provides a powerful data reference and effective methods for the key abilities and professional development of primary music education. The research will also provide better experimental methods and research models for the key competencies and professional development of teachers in other disciplines.

In addition to normal students, music AI can also effectively support students with learning disabilities to participate in classroom teaching, overcome the defects of traditional education methods, and achieve inclusive teaching. Della et al. [39] explores the impact of using AI in music education on the learning process of students with learning disabilities. Through the auditory and motor systems involved in music, students can become more independent and achieve learning goals in this situation. Students with learning disabilities are not people who don't want to learn or aren't committed enough. Not all students have a disability, but each student has a specific condition that may involve different skills at different levels. Teachers can use different approaches to meet the learning needs of all students and can use compensation tools to support students, including any technology that enhances, maintains, or improves the abilities of students affected by any type of disability (and anyone indirectly affected). Artificial music intelligence systems provide an opportunity that should be more thoroughly integrated into pedagogy, including formal and informal learning environments, teachers and their methods, available resources and activities undertaken by students. Instead of entrusting problem solving to AI-based technology, teachers must monitor dyslexic students throughout the learning process. The development of technology requires teachers to have innovative training that keeps pace with the times and can lead students in the learning process. At present, some AI technologies have begun to be applied to specific disabled groups, bringing benefits to their learning. The School of Special Education of Beijing United University uses the iFLYTEK voice transcription system to teach students with hearing disabilities. A team from the Department of Computer Science at Oxford University has developed a new AI system called LipNet to help people with hearing disabilities read lips.

## 2.3. Application to Online Teaching, Online Assistance, AI Sparring

As an important driving force of the new round of scientific and industrial revolution, AI is profoundly changing the way people live, work, and learn in education. The appli-

cation of AI technology in teaching will effectively improve the quality and efficiency of education, from classroom teaching to course guidance, from AI examinations to college entrance planning.

In the traditional teaching method, teachers mainly impart knowledge to students through dictation and PPT, which lacks interest and interaction between teachers and students, and the teaching process is boring. With the arrival of the 5G era, the technological standards have broken down the barriers to acquiring knowledge, and the quality of online education has also been improved, which can meet more personalized education needs [40].

Hua et al. [41] combine the multi-user detection algorithm of artificial intelligence to provide a good online design example for online music education, the conclusion analysis shows that the music online education system based on the SCMA system multiuser detection algorithm and artificial intelligence designed in this paper can significantly improve the audience's music learning efficiency and has obvious benefits to the student group. The system module involves basic information management, student music assignments, online courses, and other levels, providing an excellent educational system design example for music online education. The combination of AI and system has a positive impact on the future sustainable development of online music education. Through the application of online teaching of music, teachers and students get enough learning. With the aid of AI technology, the system analysis and design method are used to analyze and design a functional system of music teaching. As shown in Figure 2, when the user operating system enters data or selects options, it can be submitted by the system to the server. The physical structure of the system is connected to the physical network of the system and associated with the user terminal. Each user can connect to the mobile communication network through a mobile phone, query information through the desktop system or access the background. Students can complete music learning online through mobile phones. The system module involves basic information management, students' music homework, network courses and so on. It provides an excellent example of education system design for music network education.

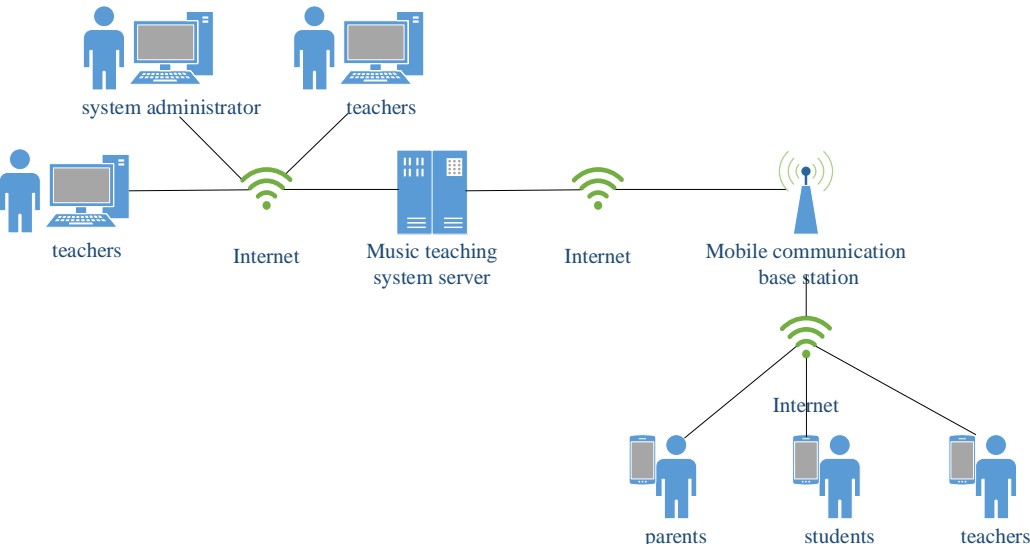

**Figure 2.** Schematic diagram of physical system structure.

Dai et al. [42] proposed and improved the "7 − 7" teaching mode based on artificial intelligence on the basis of the "4 + 3" mode of traditional classroom, which fully demonstrates the characteristics of the teaching mode under AI. The "4 + 3" model, that is, the four-operation links of teachers (lesson preparation, teaching, assignment, and evaluation) and the three learning links of students (preview, listening, and completing homework). The "7 + 7" model, that is, in smart teaching, teachers' "teaching" has become seven steps (resource release, goal setting, sensory introduction, task distribution, guidance and expla-

nation, detection and evaluation, and extension and push), and students' "learning" has also become seven steps (independent preview, learning expectation, situational experience, cooperative learning, onstage explanation, consolidation of quiz, and breakthrough points), and the interaction between teachers and students is more vivid and rich. The model of "7(medium) + 2(excellent) + 1(low)" is introduced to analyze and judge the learning situation of students in a class in a certain region. As shown in Figure 3, simple classifications combined with big data can help teachers make basic judgments on students, adopt reasonable teaching strategies and carry out classroom teaching design for all students. The teaching mode based on AI is more student-centered and focuses on the interaction between education and learning. It does not consider the single element of education or learning in the teaching process, but the complete cycle mode based on pre-class, in-class and after-class. It uses big data, internet of things, mobile internet, AI and other new generations of information technology to build a set of scientific, intelligent music teaching design models. Wisdom teaching provides reference for the whole process before, during and after class, helps guide teachers to better carry out wisdom teaching, helps students to explore cooperative autonomous learning, and promotes the wisdom transformation of teaching methods and learning methods to a certain extent. Music classroom teaching becomes more targeted and effective.

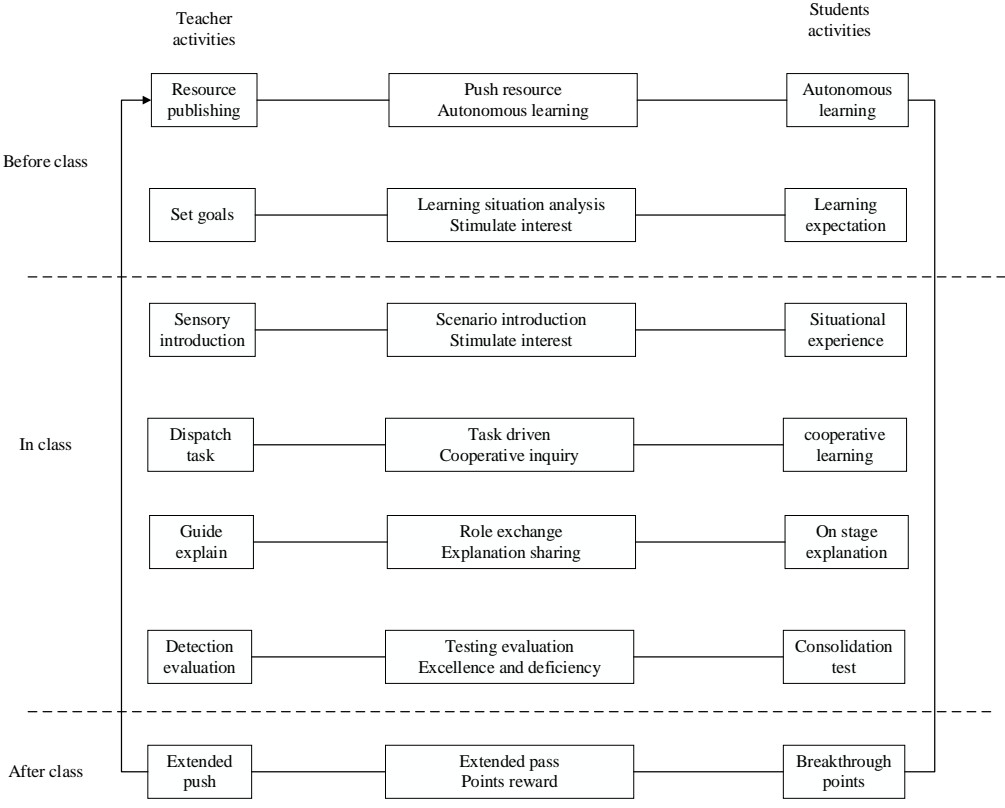

**Figure 3.** "7 + 7" mode of intelligent music teaching.

At present, some online assisted teaching software has been put on the market. In China, the Little Leaf Piano application can identify and mark the wrong tone and wrong rhythm in real-time during piano practice, through the millisecond level of artificial intelligence piano sound recognition technology and the billion level database, with high recognition accuracy. In India, a platform such as Artium Academy, provides an AI-enabled music learning experience. Learners can track and improve their learning based on AI's immediate feedback and can perform regularly online in the Artium community.

*2.4. Application to Autonomous Teaching*

For students majoring in music education, autonomous teaching using AI in the classroom is a huge challenge, because music is a field that requires students and teachers to constantly ask questions and discuss innovation. From original music education to innovative music education, teaching methods are combined with information technology to enhance the effect of education. In order for students to have a better educational experience, effective discussion and interaction between teachers and students will facilitate teachers to better understand students and help students make progress. Music teachers can continue to develop effective assessment methods, follow different methods, and evaluate students to continuously improve music knowledge.

Wei et al. [43] proposed a music education method based on AI, they thought that AI can make more optimized environments and professional music classes so that teachers and students can make the most of this and ensure smooth improvement in the network's teaching model. With the development of modern information technology, music education continues to improve. The use of AI in music education has broken the traditional mode of music teaching, greatly improving the level of music teaching and music education teaching mode. The online teaching platform for music majors based on AI technology can provide a more optimized environment and more professional music courses, so that teachers and students can make full use of this and ensure the smooth improvement of the online teaching mode. In addition, the evaluation method is described in detail in the system framework to support the development of music education. By choosing the AI system instead of other machine learning methods, both teachers and students can obtain sufficient benefits and ensure the effective improvement of the network teaching mode. Music teachers are increasingly using technology to develop new student engagement strategies. Acoustic tools and pencils used to be the primary training tools. Now students can use technology like the iPad and educational apps for creative music learning. The proposed music education and teaching based on AI techniques enhanced music education in music education management and proved a deep-level AI implementation could enable management services to be intelligent and informatized.

## 3. Development Significance and Prospects of AI and Music Education

With the advent of the era of AI, the form of education has improved with the age of network information, the singleness of school music teaching has improved, students' interest in music has been stimulated, learning efficiency has improved, and the music education model in the age of network information has been made more perfect and developed in deeper directions. On the basis of completing basic teaching, AI can also perfect each stage, so that the education concept can delve deeper into society [44–49].

In summary, with the continuous development of AI, AI has been widely popularized and penetrated the field of music education, realizing the integration and interaction between music and modern science and technology, and greatly promoting the development of the music education industry [50–52]. The combination of AI and music education is the general trend. In the future, no matter what kinds of intelligent equipment and virtual technologies, they will be more and more applied in education. The emergence of all kinds of intelligent tools will also promote the improvement of students' learning efficiency and quality. In order to assist teachers to complete the course arrangement more effectively and accurately, the prospect of introducing AI into the classroom is very broad. It can provide teachers with an auxiliary tool and pay more attention to teaching in accordance with their aptitude. Therefore, in the development of music education, we need to uphold innovative ideas, deepen the effective understanding of AI in the music education industry, strengthen the professional application of AI in music education, closely follow the development trend of AI, and promote the long-term healthy and sustainable development of the music education industry [53–55].

**Author Contributions:** Conceptualization, X.Y. and K.W.; methodology, N.M.; software, N.M.; formal analysis, L.W.; investigation, L.Z.; writing—original draft preparation, X.Y.; writing—review and editing, N.M.; visualization, L.Z.; supervision, K.W.; project administration, L.W.; funding acquisition, K.W. The named authors have substantially contributed to conducting the underlying research and drafting this manuscript. All authors have read and agreed to the published version of the manuscript.

**Funding:** This work was supported by the Shandong University Youth Innovation Team Development Plan (No. 2021RW012), National Social Science Foundation Art Program (20CD173), the Youth Fund of Shandong Province Natural Science Foundation (No. ZR2020QE212), Key Projects of Shandong Province Natural Science Foundation (No. ZR2020KF020), the Guangdong Provincial Key Lab of Green Chemical Product Technology (No. GC 202111), Zhejiang Province Natural Science Foundation (No. LY22E070007) and National Natural Science Foundation of China (No. 52007170).

**Data Availability Statement:** The data and materials used to support the findings of this study are available from the corresponding author upon request.

**Conflicts of Interest:** The authors declare that they have no known competing financial interest or personal relationship that could have appeared to influence the work reported in this paper.

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
