# Peer review of "Developments and Applications of Artificial Intelligence in Music Education"

_technologies, doi:10.3390/technologies11020042_

Round 1

Reviewer 1 Report (Previous Reviewer 4)

The response regarding hyper parameter search is still not adequate (Point 3). Please provide information about model performances if other hyper-parameters are used. Is the utilization of other optimizers considered? What about other activation functions? Please elaborate.

Author Response

Reviewer 2 Report (Previous Reviewer 2)

The article has improved, although it could delve deeper into the subject, it can be considered valid for publication. I beg in future publications to put more effort in the study of all the applications that exist in the market.

Round 2

Reviewer 1 Report (Previous Reviewer 4)

The manuscript can be accepted

This manuscript is a resubmission of an earlier submission. The following is a list of the peer review reports and author responses from that submission.

Round 1

Reviewer 1 Report

1. Guo et al. [16] and Liu et al. ----

---> A more detailed explanation of the research results on the function, role and contribution of AI is needed.

2. The paragraph below can be shortened.

As an important driving force of the new round of scientific and industrial revolusion ~~~ ~~ passively receiving knowledge in the traditional way.

3. Hua et al.[19] combine the multi-user detection algorithm ~~~

--> What are the main key ideas of thre reference? Please do not include the introductory part of the paper, but briefly summarize the research results in terms of quality and quantity.

4. Dai et al. ~~~

--> What are   "7-7",  "4+3"?

5.  Sentences similar to the following paragraphs are found before and repeated throughout  the paper. Therefore, it is recommended to erase or summarize.

At present, the future trend of online music education is to combine algorithm and 205 AI. The rapid development of AI has given people a new understanding of online educa- 206 tion, which means that in the future, online education will be more attractive than offline 207 teaching and will be widely accepted by people. With the continuous development of 208 online education, music education continues to move online. At present, although most 209 parents spend a lot of time and energy on helping and supervising their children's music 210 learning, they suffer from a lack of learning foundation and cannot correct the wrong 211 grammar during learning. Therefore, AI music training software is used as an auxiliary 212 tool for children to learn music. There are many kinds of AI music training software on 213 the market, with different functions and levels, and different experiences and effects. Little 214 star as a music training software, highlighting the strengthening of the two aspects of 215 spectrum recognition and evaluation functions, children easier to operate. When practic- 216 ing new tune, little star AI partners will remind the children of what they need to pay 217 attention to in every piece of music. According to the subject of practice to play the situa- 218 tion marked part, let the children have targeted practice. Each play has scores and evalu- 219 ation report, can let the children on their own have a comprehensive cognition practice 220 achievement, let the child can faster to master knowledge of music, playing in practice. AI 221 is widely used in various fields, profoundly changing people's lives and learning, and the 222 application of AI technology in the field of education is becoming increasingly mature, 223 which further helps the online promotion of intelligent software in music education.

6.  Wei et al.[21] ~~~

-->  What are the main key ideas of thre reference? Please do not include the introductory part of the paper, but briefly summarize the research results in terms of function, quality and quantity.

Reviewer 2 Report

It is very interesting to study the application of AI in musical intelligence. However, in this article the subject is not covered in sufficient depth. The number of references (21) is very low, given the large number of AI works and applications that currently exist. Not enough examples of applications and uses are given. For example, help for people with disabilities is only supported by a referral, which is not significant.

The article needs a major revision. Improve the structure, explain why its content and above all, present clear conclusions. The review of the state of the art is fundamental, a better vision is needed of what and how AI is used in music teaching and its comparison with other types of teaching.

Reviewer 3 Report

The paper deals with the use of Artificial Intelligence in music education. The novelty of the paper is not clear. This aspect should be reported in the abstract and in the introduction. Also the structure of the paper should be reported at the end of the section 1 to improve the output of the paper. Furthermore, several other papers are not cited such as:

Zhang, J., & Wan, J. (2020, April). A summary of the application of artificial intelligence in music education. In International Conference on education, economics and information management (ICEEIM 2019) (pp. 42-44). Atlantis Press.

Yuan, S. (2020, April). Application and study of musical artificial intelligence in music education field. In Journal of Physics: Conference Series (Vol. 1533, No. 3, p. 032033). IOP Publishing.

Jiang, Q. (2022). Application of Artificial Intelligence Technology in Music Education Supported by Wireless Network. Mathematical Problems in Engineering2022.

Reviewer 4 Report

This paper presents a review of AI applications in music education. I belive that such research has publication potential, but some major issues are present.

1. The number of cited references must be expanded to at least 40-50.

2. What about the PRISMA statement during the article colection?

3. What is the main novelty of this review paper? How it can be compared to other similar review papers?

4. There is no elegant comparison that summed up all reviewed papers. 

Round 2

Reviewer 1 Report

The author did not take the time to fully respond to the reviewer's comments.

This paper does not appear to suggest anything new due to the nature of the review paper. And overall, there are many overlapping reviews, so it would be good to organize those parts, and even if it is a review paper, it seems necessary to include the key contents of each existing study and make an effort to compare and analyze existing studies using tables.

Reviewer 2 Report

The article needs further revision, as the title suggests that artificial intelligence and music systems will be reviewed in depth. The changes made and the increase in the bibliography are not sufficient.

More comparisons, examples, etc. are needed.

Reviewer 3 Report

The authors didn' address my main issue on novelty. What is the novelty? is a review? if this is the case, this should be highlighted in the Title, abstract and introduction.

Reviewer 4 Report

My comments are addresed